# Women with Schizophrenia Have Difficulty Maintaining Healthy Diets for Themselves and Their Children: A Narrative Review

**DOI:** 10.3390/bs13120967

**Published:** 2023-11-24

**Authors:** Mary V. Seeman

**Affiliations:** Department of Psychiatry, University of Toronto, Toronto, ON M5S 3G4, Canada; mary.seeman@utoronto.ca; Tel.: +1-416-486-3456

**Keywords:** schizophrenia, nutritional deficiencies, women, antipsychotics, poverty

## Abstract

Severe psychiatric disorders such as schizophrenia are inevitably linked to unemployment, meagre per capita income, and residence in disadvantaged, poorly resourced neighbourhoods. This means difficult access to healthy food and is particularly problematic for pregnant women and mothers with children to feed. The necessity of taking antipsychotic drugs is an additional barrier to healthy eating because these drugs are associated with serious cognitive, psychological, behavioural, and metabolic sequelae. Being ill with psychosis makes it extremely difficult to maintain a healthy diet; nutritional deficiencies result, as do medical complications. The results of present literature review confirm the gravity of the problem and suggest a number of potentially useful clinical interventions.

## 1. Introduction

Clinical psychiatrists seldom ask schizophrenia patients what they eat. The evidence appears to indicate that they probably should. What is widely known is that too many patients with schizophrenia, especially women, suffer from obesity, partly due to the relative affordability of high calorie food, partly due to lifestyle, and partly resulting from treatment with antipsychotic drugs [1]. What is perhaps less well-known is that anorexia can also occur in the context of schizophrenia [2,3,4]. Healthy eating, as defined by the World Health Organization in 2020 [5], refers not only to quantity, but also to the variety, balance, and safety of a person’s regular, day-in-day-out diet. Healthy eating is difficult to maintain for individuals diagnosed with schizophrenia and related disorders. In general, patients with any severe mental illness (SMI) eat high energy (sugar and saturated fat), relatively little fibre and fruit, and food that contains more salt than that of healthy controls [6,7]. The SMI diet is best characterized as high in easily accessible ready-to-eat food and low in fruit and vegetables [8,9]. A review of 822 studies of diet and psychosis in 2020 [10] found poor quality dietary patterns in individuals with psychosis, high intake of refined carbohydrates and total fat, and low intake of fibre, ω-3 and ω-6 fatty acids, vegetables, fruit, and important-to-health vitamins and minerals (vitamin B_12_ and B_6_, folate, vitamin C, zinc, and selenium). On average, individuals with SMI also exhibit considerably less knowledge about nutrition than their healthy peers [11]. Because this suggests that routine checks on patients’ food intake may need to become a routine component of psychiatric practice, I undertook a narrative review of the literature on (A) the causes of poor dietary habits in schizophrenia-related disorders, (B) the presence of special dietary needs, (C) the general health consequences of poor nutrition in this population, and (D) potential interventions that show promise in ameliorating the situation.

## 2. Materials and Methods

This narrative review of the difficulties that women with schizophrenia face in maintaining healthy nutrition standards for themselves and their children is not a systematic review of the extant literature on nutrition and schizophrenia, which is a far broader topic. In order to find the most pertinent (a subjective term) evidence useful for clinicians, PubMed and Google Scholar titles and abstracts were searched over the last ten years, using the following search terms: (A) Diet OR Nutrition OR Food AND Schizophrenia OR Severe Mental Illness OR Psychosis AND Causes; (B) Diet AND Needs AND Schizophrenia OR Severe Mental Illness OR Psychosis; (C) Schizophrenia OR Severe Mental Illness OR Psychosis AND Consequences AND Nutrition OR Diet OR Food; (D) Schizophrenia OR Severe Mental Illness OR Psychosis AND Intervention OR Treatment OR Prevention AND Nutrition AND Deficiency OR Lack. Primary research and reviews were included; papers in languages other than English were excluded. The reference lists of the full articles were subsequently searched for papers published prior to 2013 and considered by the author to be clinically relevant. The final selection of papers to include in this narrative review was admittedly subjective, based on clarity of writing, clear methodology, recency, and novelty of conclusions.

## 3. Results

### 3.1. Causes for Poor Diet in Schizophrenia

#### 3.1.1. Poverty

Healthy diets are costly, which is a major reason why they are difficult to maintain by individuals with schizophrenia, most of whom are unemployed and living on disability or old age pensions in deprived urban environments. Majuri et al. [12] in 2023 published a study of the 30-year employment trajectory patterns of patients with schizophrenia based on the Northern Finland Birth Cohort of 1966 (number with schizophrenia = 62, ages 16–45). As has been shown in earlier work, most individuals with schizophrenia were not in the labour market and relied, instead, on a government pension. A 7-year study (2006–2013) from Sweden [13] found employment to be rare (10%) among people with schizophrenia; over 80% were being supported on a disability pension. Writing about the finances of people with major psychiatric disorders, Eliason [14] summarizes the situation: “even in an advanced welfare state, people with SMI (severe mental illness), especially those with schizophrenia, have an extremely weak position on the labour market and an equally difficult financial situation”.

There are reasons beyond low income for poor nutrition in this population. Low-income urban neighbourhoods, where people with schizophrenia tend to live, do not provide easy access to healthy food. In other words, supermarkets that offer fresh, good quality food at affordable prices are missing from these neighbourhoods [15,16]. Accessible grocery stores sell mostly processed food. The residential areas of individuals with schizophrenia are “food deserts”, which means that one has to travel a fair distance to purchase healthy food, which, in turn, incurs unaffordable transportation costs [15], especially so for house-bound mothers.

#### 3.1.2. Schizophrenia Symptoms

The comprehensive review of dietary habits associated with psychosis by Aucoin et al. [10] acknowledges that socio-economic status, but also psychotic symptoms, are responsible for the poor nutrition seen in this population. Positive symptoms (hallucinations, delusions), negative symptoms (apathy, social isolation, anhedonia), cognitive deficiencies (attention, short term memory), and mood problems (depression, anxiety) conspire in schizophrenia to result in a reduced interest in food (anhedonia) and inhibit all motivation to prepare food (avolition). Furthermore, social isolation tends to make eating more of a chore than a pleasure. It is known that food is best enjoyed in company; eating alone undermines appetite and makes shopping and preparing meals feel pointless [17,18]. Findings from a review of 42 studies suggest that, relative to eating alone, eating with others significantly increases food intake but that this is moderated by the identity of a person’s meal companions. Human beings tend to imitate the food choices and preferences of co-eaters [19]. Apathy, a common feature of schizophrenia [20], has been shown in one study to impede living skills, measured using the Independent Living Skills Survey (ILSS) [21], which includes items relevant to food preparation. Semkovska et al. [22] examined three activities (choosing a menu, shopping for the ingredients, cooking a meal) performed by 27 patients with schizophrenia and 27 controls. Patients with schizophrenia committed significantly more omissions when choosing the menu, made more errors during the shopping task, and also during the cooking task. The degree of error was found to correlate significantly with the severity of the patient’s negative symptoms [22]. Cognitive problems have not been studied in relation to schizophrenia and food tasks, but undoubtedly contribute to the inability to budget for food and to prepare healthy meals, in the same way as they have been shown to do in Alzheimer’s disease [23].

The positive symptoms, delusions and hallucinations - e.g., hearing commands to desist from certain foods or avoid certain grocery stores for fear of danger - may deprive patients with schizophrenia of specific foods. Occasional case reports attest to this possibility—these are reports of Wernicke’s thiamine deficiency and encephalopathy associated with schizophrenia (in the absence of alcohol consumption) [23,24,25,26,27,28]. Alcohol addiction may, nonetheless, in many cases of schizophrenia, be the main cause of Wernicke’s syndrome [29].

#### 3.1.3. Circadian Rhythms in Schizophrenia

Disruption of the circadian rhythm can interfere with mealtimes and interfere with regular eating. Both animal and human data point to sleep and circadian rhythm disturbance in schizophrenia, mediated perhaps by increasing dopamine release and sensitivity [30]. This affects sleep patterns such as day/night reversal, which, in turn, throw eating patterns into disarray. [31]. It has been suggested that a relationship exists between light exposure and mealtime regularity and that lack of light can have negative effects on metabolic health [32]. Being unemployed, as most persons with schizophrenia are, means few daytime obligations, which, in turn, means that sleeping by day and staying awake at night (night/day reversal) is a frequent mode of living for this population because it avoids social encounters (which many with this disorder find distressing). Avoiding social contact, however, encourages shopping in fast food stores and, thereby, negatively impacts nutrition [33].

#### 3.1.4. Substance Excess in Schizophrenia

Individuals who are socially isolated are particularly vulnerable to alcohol and drug dependency [34], while chronic substance abuse is known to be positively associated with food deficiencies and malnutrition. Ross et al. [35] conducted a study of 67 patients in an alcohol detoxification unit and assessed their nutritional status through questionnaires and blood tests. Twenty-four percent of the study participants qualified for a diagnosis of mild to moderate malnutrition, a percentage that the study authors considered an underestimate. Half the participants were found deficient in iron or vitamins. Alcohol consumption, especially in males, is highly prevalent in the schizophrenia population [36] and probably plays an important role in unhealthy eating.

#### 3.1.5. Living Circumstances

When not hospitalized, individuals with schizophrenia live either with family or they live alone or in communal living quarters. The living arrangements of over 1446 schizophrenia patients from 57 sites throughout the United States over a one-year period were characterized by Tsai et al. [37]. At baseline, 46% were living with family members, 5% were living with nonrelatives, and 18% were living independently. Seventeen percent were housed in an institution, and 14% lived in unstable housing. In another study of 1825 participants with psychotic disorders, one half lived in public or private rented housing (48.6%) and 22.7% were waiting in line for public housing. Thirteen percent lived in their own home. One in 20 study participants was homeless at the time of the study; almost 13% had been homeless at some point within the previous 12 months [38]. Because of the close correlation between healthy eating and safe, stable housing [39], according to the results of these studies, between 12% and 14% of individuals with a psychotic disorder are precariously nourished [40,41]. This does not mean they are not eating, but that they are eating on the fly, without regard to food content.

#### 3.1.6. Antipsychotics

Antipsychotic, antidepressant, and mood-stabilising medications are all associated with substantial weight gain, which can result, over time, in obesity and cardiometabolic abnormalities. Elman et al. [42] have suggested that the mesolimbic hyperdopaminergic state that is hypothesized to exist in schizophrenia makes food taste bland unless extra salt or sugar is added. This, plus cognitive deficiencies, plus symptoms of psychosis, plus limited resources, plus alcohol help account for many of the unhealthy eating habits seen in schizophrenia. When treatment with antipsychotics is added to the mix, it lowers dopamine levels so that positive symptoms diminish, but appetite is increased. The result is overeating of unhealthy food. The findings of a study by Blouin et al. [43] are consistent with this view. Individuals taking second generation antipsychotic medication report increased appetite, decreased satiety levels and, in addition, increased cravings for sweet foods and beverages. Two more recent reviews of this topic confirm these results [44,45], but some antipsychotics do have less propensity for metabolic side effects than others.

Mukherjee et al. [44] underscores the importance of a drug’s individual receptor binding profile to its impact on nutrition. Affinity for dopamine D2, serotonin 5HT2c (one of the 14 members of the 5-hydroxytryptamine receptor family), and histamine H1 receptors plays a critical role in a drug’s interaction with appetite regulation pathways in the brain.

Second generation antipsychotics are known to induce more metabolic ill effects than earlier drugs. In 2021, Rognoni et al. [46] conducted a review of 79 papers to determine the worst among the drug offenders, excluding clozapine, which was not part of their study. Olanzapine showed the largest body mass index (BMI) increase; paliperidone showed the highest increase in total cholesterol; quetiapine once daily dosing showed the highest decrease in fasting glucose and the highest increase in systolic and diastolic blood pressure. Lurasidone showed a decrease in total cholesterol (−8.01 mg/dL) and triglycerides (−5.33 mg/dL) [46]. In another recent review, olanzapine and clozapine were reported as the antipsychotic drugs causing the greatest weight gain [47]. These drugs are, however, sedative, so that patients who, on other drugs, were snacking at night paradoxically lose weight on olanzapine and clozapine because they sleep more soundly!

Antagonistic effects on muscarinic and histaminergic receptors contribute to increased appetite. Additional variables are the drug’s effect on gut microbiota and the brain areas that regulate food intake. All antipsychotics change the balance of gut bacteria [48] and it may be this indirect route that leads to various metabolic disturbances. The research groups of both Mukherjee et al. and Sankaranarayanan et al. [44,45] conclude their reviews of the effects of 2nd generation antipsychotics on disordered eating in schizophrenia by detailing the consequences of their use on a number of dietary and nutritional parameters, namely hunger, food cravings and addiction, dietary restraint and disinhibition, satiety, caloric intake, eating habits, and dietary composition.

### 3.2. Special Dietary Needs

#### 3.2.1. Schizophrenia and Diabetes

There are circumstances and health conditions that make good nutrition especially important. Comorbidity between schizophrenia and diabetes mellitus Type 2 (DM) is one such condition and it is highly prevalent among individuals with schizophrenia. The rate of DM is two to three times higher in schizophrenia than in the general population [49]. It has been hypothesized that the two disorders share common susceptibility genes [49]. Moreover, the antipsychotic drug treatment for schizophrenia is known to induce weight gain and metabolic abnormalities such as DM. Preventing diabetes and controlling diabetes means paying careful attention to what one eats in the context of a well-ordered lifestyle. This is difficult for persons with schizophrenia, in part because of cognitive defects, in part because of socioeconomic disadvantage. Focus groups held with individuals comorbid for DM and mental illness (not only schizophrenia) have uncovered many subjectively reported challenges in this population, any of which can lead to poor DM self-care [50]. Qualitative interviews with psychiatric patients, family members, and primary care staff yielded similar results [51]. A wide range of factors are known to influence DM self-management, which refers not only to dietary content and eating routines but also to motivation, knowledge, and skills, as well as to social support from family and friends, all of which are lacking in most patients with schizophrenia [52].

#### 3.2.2. Schizophrenia and Pregnancy

Pregnancy incurs special nutritional needs in women. Although this has not been extensively studied, vitamin deficiency (especially Vitamin D deficiency) and mineral deficiencies (zinc and iron) have been reported and have been positively associated with intrauterine growth restriction in infants born to women with SMI [53], whereas because of antipsychotic-induced obesity, pregnant women with schizophrenia need larger than average doses of routine prenatal folic acid supplements, in one study, folic acid was only being taken by 25% of pregnant schizophrenia women [54]. The same paper [54] reported iron deficiency in 73% of the women. A study of 38 pregnant women with SMI found high rates of obesity, with 82% showing gestational weight gain above recommendations. Almost one-third of the women failed to meet any of the Five Food Group serving recommendations for pregnancy and reported above-recommended levels in their diet of processed food (19%) and sugar snacks (51%) [54]. The likelihood of pregnant women with schizophrenia treated with antipsychotic medication developing gestational DM has been reported as 5–7 times that of women with schizophrenia not taking these drugs [55]. First generation antipsychotics were seen as the worst culprits [55]. Because of lack of education, high levels of misinformation, and the socio-economic status of women in low-income countries, malnutrition during pregnancy remains an acute health problem in many regions of the world [56].

#### 3.2.3. Schizophrenia and Motherhood

Qualitative studies have addressed healthy food access and appropriate nutrition in women with schizophrenia who are also mothers, most often single mothers. Lack of time (for shopping and cooking) and difficulties in finding transportation to and from food stores were reported by mothers in large urban centres to be major constraints to the provision of adequate nutrition for their children [15,57]. Low-income mothers are known to often prioritize food pricing and shelf-stability when buying food, thereby sacrificing food quality [58]. Many environments, but especially poor environments, are plagued according to mothers, with street vendors selling non-nutritious high caloric food that is a magnet for children [58]. With respect to mothers’ own nutritional needs, during periods of food insecurity, mothers routinely compromise their own food needs in order to better feed their children [59]. This makes mothers with schizophrenia a particularly nutrition-vulnerable group.

#### 3.2.4. Schizophrenia and Old Age

An even more nutrition-vulnerable group are the elderly, where the prevalence of severe malnutrition is reportedly very high [60] and has been attributed to “the nine ds”: dentition, dysgeusia, dysphasia, diarrhoea, depression, disease, dementia, dysfunction, and drugs [61]. A recent study [62] has highlighted the functional impairments seen in older (mean age = 64) patients with schizophrenia. Because of physical, psychological, and cognitive morbidities, many participants in this study were unable to carry out their daily activities. In this study, 16% were underweight and 28% were overweight or obese. Authors of a study from Iran who recruited chronic psychiatric hospital patients aged 60–75 [63] reported that almost 60% were at risk for malnutrition. In a study of schizophrenia patients in the community (all ages), that percentage was 45% [64].

### 3.3. Health Consequences of Poor Nutrition

What one eats predicts the occurrence of many health conditions, including, among others, atherosclerosis, obesity, diabetes, hypertension, dyslipidaemia, anaemia, and sarcopenia [65,66,67,68]. These studies show that anaemia can result from an inadequate diet even in the absence of blood loss. They refer to randomized controlled trials that show that intake of protein is critical for preservation of muscle strength, although the optimal dose and type of protein remains unknown. They demonstrate that inadequate nutrition, in addition to factors such as low amounts of physical activity and lack of hypertension control, are significant triggers of both cardiovascular and metabolic disease. Of specific importance to women, nutrition during pregnancy has been shown to importantly affect the health of the foetus [69].

Nutrition influences opportunities for employment; at work, it affects both productivity and performance [70]. Nutrition also affects the speed of the aging process and is a good predictor of longevity [71].

### 3.4. Interventions for Poor Nutrition in Schizophrenia

Interventions start with recognition of the problem and individual risk assessment. After a comprehensive review of disordered eating in schizophrenia, Sankaranarayanan et al. [45] conclude that clinicians need training in optimal ways to question severely ill psychiatric patients about their eating habits, assess their needs and manage their nutritional deficiencies. Smith et al. [72] also address the need to question patients with SMI about their ease of access to healthy food.

Using assessment measures such as the Geriatric Nutritional Risk Index (GNRI) and Onodera’s Prognostic Nutritional Index (OPNI) for long-term outcomes over a median period of 408 days, Tsai et al. [73] found that the GNRI predicted falls in hospitalized schizophrenia patients and the OPNI predicted both falls and infections. A scoping review of 17 nutrition screening methods for adults with severe mental disorder was recently identified [74]. Three had been validated specifically for this population.

After an appropriate assessment, which should ideally include a home visit, specific interventions are in order. Classes in meal planning, cooking, and eating behaviours have shown good results. Clark et al. [75] assessed 18 individuals with severe mental illnesses pre and post a 6-week nutrition education cooking class. They found that self-reported efficacy in cooking and grocery shopping skills improved and patients reported significant increases in calcium, vitamin D, grains, and fruit intake. Garcia and Privott [76], in a N = 1 qualitative study, discussed challenges and barriers to taking part in cooking classes in their description of one individual’s transition to independent community living after long-term hospitalization with SMI. They attribute recovery to the person’s sustained participation in meal preparation and cooking groups while in hospital.

Positive results with respect to healthy eating have been seen with both cognitive behavioural therapy and pharmacological intervention. Naltrexone prevents antipsychotic-induced weight gain by reducing food cravings [77]. The new anti-obesity drugs may, in the near future, be of great help to this population.

Providing patients with opportunities to socialize is another way to expose them to healthier eating habits. Successful online intervention has also been reported [78]. Food support (financed and delivered via a number of diverse ways) has been the way that many parts of the world utilize to help maintain adequate nutrition in individuals with schizophrenia and related disorders [79].

## 4. Discussion

Nutrition is a very large field. For instance, this review does not cover the possibility that prenatal nutritional deficiency in mothers has been suspected of increasing the risk of schizophrenia in offspring [80,81]. Animal research has bolstered this hypothesis. For instance, Chen et al. [82] compared hippocampus and prefrontal cortex transcriptomes in rats with and without prenatal nutritional deficiency. They identified genes involved in synaptic development, neuronal projection, cognitive function, and learning function in rat hippocampus (none in the prefrontal cortex) that responded to maternal prenatal nutritional deficiency; four of the genes were associated with schizophrenia. This work suggests that effective prenatal care of women with schizophrenia is of vital importance to their offspring.

After years of relative medical neglect, nutrition and its effect on mental health has become a topic of active current research. This research field, however, has tended to focus on aspects of diet relating to depression. Examples are plans for interventional studies for dietary modification and nutraceuticals based on personalized biomarkers [83,84]. In the study of schizophrenia, older research advocated vitamins for schizophrenia [85] and dietary factors such as gluten have for decades been seriously investigated as potential schizophrenia causes [86]. Today, the schizophrenia and nutrition field is best summarized by the Aucoin et al. [10] review. Their review does not address the hypothesis that food deficiencies can cause psychosis but does conclude, on the basis of literature, that there is demonstrated benefit for people with psychosis eating a diet rich in vitamins and minerals (vitamin B_12_ and B_6_, folate, and zinc) and amino acids (serine, lysine, glycine, and tryptophan). The authors do acknowledge, however, that financial barriers, inadequate education and poor budgeting skills, as well as lack of motivation, make it very difficult for individuals with psychosis to achieve and sustain healthy diets, the issues that have been the focus of the present review. Aucoin and colleagues [10] advocate for the integration of nutrition experts into mental health teams.

This review has underscored the degree of poor nutrition seen in a sizable minority of patients with severe mental illness and has attributed it to a host of factors working together: economic status, psychotic symptoms, lifestyle effects, social isolation, and drug-induced effects. There is general agreement that nutritional deficiencies often result in a variety of serious medical disorders and that their prevention needs to become an inherent component of comprehensive schizophrenia treatment.

## 5. Conclusions

This review discusses poor nutrition among individuals diagnosed with schizophrenia, emphasizing the significance of factors such as poverty, social isolation, lifestyle and the use of antipsychotic medications. The emphasis is on women because they are the ones usually responsible for feeding their households. The causes of inadequate dietary habits among individuals with schizophrenia are diverse and include specific symptoms associated with this disorder, disturbances in circadian rhythms, overuse of substances, and challenging living conditions. Poor nutrition results in a host of medical conditions such as atherosclerosis, obesity, diabetes mellitus, hypertension, anaemia, and sarcopenia. It incurs risks to the foetus in pregnant women with schizophrenia. The review also explores potential interventions, such as nutrition education, cognitive-behavioural therapy, pharmacological approaches, social support, and online interventions. It underscores the need for clinicians to systematically assess and address nutritional deficiencies as an integral part of comprehensive care for individuals with schizophrenia.

## Data Availability

Not applicable.

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
