# Peer review of "Women with Schizophrenia Have Difficulty Maintaining Healthy Diets for Themselves and Their Children: A Narrative Review"

_behavsci, 2023, doi:10.3390/bs13120967_

Round 1
Reviewer 1 Report (Previous Reviewer 3)
Comments and Suggestions for Authors
Congratulations.
Author Response
Thank you.
Reviewer 2 Report (New Reviewer)
Comments and Suggestions for Authors
A comprehensive narrative review of an important aspect of managing women (and actually all patients) with schizophrenia, that as such, makes a contribution to the literature.
The article is fine largely as is. However, perhaps the following points are of utility:
line 23 -- typo: "...due to lifestyle..."
line 40 -- typo: "..dietary habits..."
Section 3.1.6 -- It would be interesting to know what (or if) the literature says about the rough magnitude of improvement in patients' dietary consequences by changing medication (e.g., olanzapine to lurasidone) compared to other interventions, particularly if other aspects of nutrition remain unchanged.
Section 3.2.1 -- It would be interesting to know what the literature says about intervention with glucagon-like peptide-1 receptor agonists (e.g., semaglutide), where allowed of course, particularly in patients where there is only obesity and no diagnosis of diabetes. e.g., https://pubmed.ncbi.nlm.nih.gov/37113745/
(Although perhaps this is beyond the scope of the article.)
Well written, excellent review of the subject.
Author Response
line 23 -- typo: "...due to lifestyle..." Thanks!
line 40 -- typo: "..dietary habits..." Thanks!
Section 3.1.6 -- It would be interesting to know what (or if) the literature says about the rough magnitude of improvement in patients' dietary consequences by changing medication (e.g., olanzapine to lurasidone) compared to other interventions, particularly if other aspects of nutrition remain unchanged.
I have added this.
Section 3.2.1 -- It would be interesting to know what the literature says about intervention with glucagon-like peptide-1 receptor agonists (e.g., semaglutide), where allowed of course, particularly in patients where there is only obesity and no diagnosis of diabetes. e.g., https://pubmed.ncbi.nlm.nih.gov/37113745/
(Although perhaps this is beyond the scope of the article.)
Very interesting but, I agree, beyond the scope of this article.
Reviewer 3 Report (New Reviewer)
Comments and Suggestions for Authors
Thank you for exploring an interesting and important topic. My main feedback is to improve the objectivity and rigor with which this review was performed- in its current form the individual manuscripts reviewed are only briefly described and not explored on an analytical level. This topic is extremely broad- I would suggest choosing one of the subcategories from the results section to review and taking a more comprehensive dive into all of the literature currently available in this area. You should have a table or figure describing the search strategy in a way that can be replicated and a table providing more specific detail about each article included. See the PRISMA checklist for guidance- your review doesn't have to fully transform into a systematic meta-analysis but would benefit from more formal structure. A few additional comments are listed below:
- Avoid using first person language.
- Inclusion/exclusion criteria should be used to describe an objective strategy for manuscript selection.
- Avoid directly quoting other manuscripts.
- Recommend using a method to assess quality of the studies included such as the Newcastle-Ottawa Quality Assessment.
- The author also makes many generalizations that may be anecdotal rather than evidence-based (e.g. symptoms of schizophrenia "inhibit all motivation to prepare food", night/day reversal due to unemployment, weight loss with clozapine and olanzapine, etc.
Comments on the Quality of English LanguageNo issues
Author Response
The reviewer’s points are all excellent and I agree with them all, but for another kind of paper. This particular manuscript is intended to be subjective, as my aim is to persuade my fellow clinicians to inquire about food habits when assessing and monitoring patients with schizophrenia. It is not meant as an objective literature review (AI would be much better at that) but as one clinician’s, based on long experience, appraisal of the situation.
I have removed the first person language and have admitted to subjectivity in selecting papers to cite.
Round 2
Reviewer 3 Report (New Reviewer)
Comments and Suggestions for Authors
I have reviewed the author’s response and resubmission. My recommendation would be the same (major revision) since very few changes were made. I understand the author’s point that this is a subjective/persuasive work, and while it is an interesting and important topic, I feel that I can’t recommend publication in a scientific journal in it's present form.
Author Response
I have found it impossible to address further the issues raised by the reviewer. I thank him/her for the interest shown in the paper but must acknowledge a stalemate.Round 3
Reviewer 3 Report (New Reviewer)
Comments and Suggestions for Authors
I have no changes to my previous recommendation. I agree with the author that we are at an impasse.
This manuscript is a resubmission of an earlier submission. The following is a list of the peer review reports and author responses from that submission.
Round 1
Reviewer 1 Report
Comments and Suggestions for Authors
Thank you for allowing me to review this manuscript. The manuscript delves into the issue of poor nutrition among individuals diagnosed with schizophrenia, emphasizing the significance of factors such as lifestyle and the use of antipsychotic medications. The manuscript's findings delve into the underlying causes of inadequate dietary habits among individuals with schizophrenia, which are diverse and include poverty, various schizophrenia symptoms (both positive and negative), disturbances in circadian rhythms, substance abuse, and challenging living conditions. The authors detail the adverse consequences of poor nutrition, linking it to various health conditions such as atherosclerosis, obesity, diabetes, hypertension, anemia, and sarcopenia, emphasizing the crucial role of nutrition during pregnancy. The manuscript also explores potential interventions, such as nutrition education, cognitive-behavioral therapy, pharmacological approaches, social support, and online interventions. It underscores the need for clinicians to systematically assess and address nutritional deficiencies as an integral part of comprehensive care for individuals with schizophrenia, with specific attention given to safeguarding the long-term health of women in this demographic. The author briefly touches on the bidirectional relationship between nutrition and schizophrenia, particularly the potential impact of prenatal nutrition on offspring, while acknowledging that research interest in the relationship between diet and mental health is growing. Finally, the manuscript highlights the multiple challenges individuals with severe mental illnesses such as schizophrenia face in maintaining a healthy diet due to economic, cognitive, and lifestyle factors. The manuscript asserts that addressing nutritional deficiencies should be an essential component of treating schizophrenia, with particular attention given to the long-term health of women.
Author Response
This is a lovely summary of the paper. I have borrowed from it to redo the conclusion of the revised paper. Thank you.
Reviewer 2 Report
Comments and Suggestions for Authors
The manuscript describes the Nutrition in Schizophrenia, where The author listed societal events and other parameters. The author should also list/check the nutrition values that will help to retard Schizophrenia. Schizophrenia disease model should be explained in the introduction. Schizophrenia, being a neuro-disorder, the author should find some novel results in the patient outcome where nutrition of the patients is pivotal. or the paper is of less opined in this journal.
Comments on the Quality of English LanguageThe manuscript describes the Nutrition in Schizophrenia, where The author listed societal events and other parameters. The author should also list/check the nutrition values that will help to retard Schizophrenia. Schizophrenia disease model should be explained in the introduction. Schizophrenia, being a neuro-disorder, the author should find some novel results in the patient outcome where nutrition of the patients is pivotal. or the paper is of less opined in this journal.
Author Response
All revisions are in yellow.
The manuscript describes the Nutrition in Schizophrenia, where The author listed societal events and other parameters. The author should also list/check the nutrition values that will help to retard Schizophrenia.
There are no reliable ones that I could find.
Schizophrenia disease model should be explained in the introduction. Schizophrenia, being a neuro-disorder, the author should find some novel results in the patient outcome where nutrition of the patients is pivotal. or the paper is of less opined in this journal.
I am not sure what you mean. The paper does not address the contribution of specific nutritive factors to the development of schizophrenia nor foes it address any potential curative nutritive factors. The paper addresses the fact that women with schizophrenia are too poor, too cognitively impaired, too unsupported, too apathetic, too hampered by disease and side effects to provide good nutrition for themselves and their children.
Reviewer 3 Report
Comments and Suggestions for Authors
TITLE:
This is a very short title. Please add some more words. Is this an opinion or an editorial? It is an original article or a review? If i tis a review, what kind of review? Narrative? Systematic? Scoping? (etc.) If the Author is especially worried with nutrition in women with schizophrenia that should also be clear on the title of the article. Is the author is focused in anorexia and weight loss that should also be clarified in the title.
AUTHORS
The Affiliation is too short. Please provide the Lab, the Team, the Department or Institute inside the University of Toronto.
ABSTRACT
First sentence of Abstract sounds odd to me. Please rephrase it. Please, do not use “mental illness”. This is an outdated concept. Use “psychiatric disorder” instead.
Last sentence of Abstract sounds odd. Why not “severity” instead of “gravity”?
KEYWORDS
If the Author is especially worried with anorexia and weight loss in women with schizophrenia that should have been used in the keywords.
INTRODUCTION
Pure or true anorexia nervosa is very rare. It is even more rare among patients with Schizophrenia. Anorexia nervosa is a diagnosis that can only bem ade if there is no criteria for Schizophrenia. Anorexia nervosa can happen before (as na obsessive prodrome of Schizophrenia), but not, never, after Schizophrenia diagnosis, for hierarchic diagnosis reasons. Anorexia as a symptom, in the other hand, is very common among patients with Schizophrenia, mostly secondary to gustative, olfactive, or auditory hallucinations but also as a consequence of persecutory or poisoning delusions. The relation between anorexia nervosa and Schizophrenia should be explained in more detail.
There are other vitamins that should be cited here, such as Vitamin B1, B3, B6 or B9.
METHODS
PRISMA guidelines should have been used in the review.
Please, provide a Figure with a fluxogram explainig the review methodology.
The words used in the search are way too many, and go much beyound Schizophrenia. On the other hand the Author did not used any word related with women, that seemed to be, in the Abstract an importante issue. The Author, seems a bit lost through the Article, and should not use the first person (eg “I”).
FIGURES
I would recommend at least one figure to present methodology.
RESULTS
English language is poor. Schizophrenia is mispelled in the subheading number 3.1.2.
When the author focus in Whernicke syndrome should also focus in other types of vitamin deficiencies, such as Korsakoff syndrome, Marchiafava-Bignami disease, pelagra, scurvy, etc.
Results are focused in anorexia and weight loss. That was a problem, among patients with Schizophrenia, living in the 20th century. Nowadays, in the 21st century, the bigger nutrition problem among patients with schizophrenis is hyperphagia and obesity.
I have been working with Homeless suffering with Schizophrenia, and other psychoses, for 15 years in na European second world country. Most of them suffer not from caquexy but from obesity. The Author should ackowledge that low weight is a problem among Psychotic Homeless people in the third world countries, but not among the first world countries, where there is a surplus of food and i tis much more easily to get fat than thin.
Please, explain all acronyms, such as 5HT2c, BMI, XR, etc.
Whenever writting regarding the effects of antipsychotic medication i tis of utter most importance to separate second generation from first generation and third generation. Weight gain is much more common in patients taking second generation anti-psychotics. Iatrogenic symptoms like hyperphagia, polifagia and iatrogenic metabolic syndrome should be worked through also in this section.
Whenever using the word diabetes it is importante to explain that the Author is refering to diabetes mellitus. Please, write it in italics as it is an non english, latin expression. Otherwise Readers may think the Author i salso writting about other types of diabetes, such as diabetes insipida, which is a completely diferente condition.
I found a diferente type of letter, without apparent reason from Line 267 to 272, an then again, from line 279 to 282.
TABLES
I would recommend at least one table to present results.
DISCUSSION
Very poor. First sentence makes no sense. What is the meaning of “nutrition and schizophrenia may be bidirectional”? The authors should write “nutrition problems” or something like that.
CONCLUSION
Not thah much different from Discussion. Makes no sense to keep two different sections for the Article in its present form.
GENERAL
The article seems more suitable for a journal dedicated to Psychiatry or Nutrition.
Author Response
This is a very short title. Please add some more words. Is this an opinion or an editorial? It is an original article or a review? If it is a review, what kind of review? Narrative? Systematic? Scoping? (etc.) If the Author is especially worried with nutrition in women with schizophrenia that should also be clear on the title of the article. Is the author is focused in anorexia and weight loss that should also be clarified in the title.
The title has been changed to: Women with schizophrenia have difficulty maintaining healthy diets for themselves and their children: A narrative review.
I believe that this reviewer has misunderstood the intent of this review . I have tried, in the revision to make it more clear. The best and most accurate summary is the one provided by Reviewer 1.
AUTHORS
The Affiliation is too short. Please provide the Lab, the Team, the Department or Institute inside the University of Toronto.
I have added the Department of Psychiatry
ABSTRACT
First sentence of Abstract sounds odd to me. Please rephrase it. Please, do not use “mental illness”. This is an outdated concept. Use “psychiatric disorder” instead. Last sentence of Abstract sounds odd. Why not “severity” instead of “gravity”?
I have rewritten the abstract. I actually prefer the word “gravity” in this context.
KEYWORDS
If the Author is especially worried with anorexia and weight loss in women with schizophrenia that should have been used in the keywords.
The issue is not weight loss and anorexia but maintenance of a healthy diet.
INTRODUCTION
Pure or true anorexia nervosa is very rare. It is even more rare among patients with Schizophrenia. Anorexia nervosa is a diagnosis that can only bem ade if there is no criteria for Schizophrenia. Anorexia nervosa can happen before (as na obsessive prodrome of Schizophrenia), but not, never, after Schizophrenia diagnosis, for hierarchic diagnosis reasons. Anorexia as a symptom, in the other hand, is very common among patients with Schizophrenia, mostly secondary to gustative, olfactive, or auditory hallucinations but also as a consequence of persecutory or poisoning delusions. The relation between anorexia nervosa and Schizophrenia should be explained in more detail.
Anorexia is a side issue, mentioned because it is not well known in the context of schizophrenia.
There are other vitamins that should be cited here, such as Vitamin B1, B3, B6 or B9.
This is a review so I cite what I found in the literature.
METHODS
PRISMA guidelines should have been used in the review.
It’s a narrative review
Please, provide a Figure with a fluxogram explainig the review methodology.
I find these diagrams unnecessary.
The words used in the search are way too many, and go much beyound Schizophrenia. On the other hand the Author did not used any word related with women, that seemed to be, in the Abstract an importante issue. The Author, seems a bit lost through the Article, and should not use the first person (eg “I”).
I appreciate the reviewer’’s viewpoint. The focus was not on women but on women with disorders such as schizophrenia, which lead to food inadequacy. I omitted the personal "I"
FIGURES
I would recommend at least one figure to present methodology.
I don’t believe a figure is necessary for this narrative review. All the information is in the text.
RESULTS
English language is poor. Schizophrenia is mispelled in the subheading number 3.1.2.
There was indeed a typo in the subheading. Thank you for spotting it. I could not find instances of poor English.
When the author focus in Whernicke syndrome should also focus in other types of vitamin deficiencies, such as Korsakoff syndrome, Marchiafava-Bignami disease, pelagra, scurvy, etc.
I did not find references to other dietary syndromes other than Wernicke's linked to schizophrenia in my review of the literature, though they may well exist.
Results are focused in anorexia and weight loss.
I think that is a misunderstanding.
That was a problem, among patients with Schizophrenia, living in the 20th century. Nowadays, in the 21st century, the bigger nutrition problem among patients with schizophrenis is hyperphagia and obesity.
I have been working with Homeless suffering with Schizophrenia, and other psychoses, for 15 years in na European second world country. Most of them suffer not from caquexy but from obesity. The Author should ackowledge that low weight is a problem among Psychotic Homeless people in the third world countries, but not among the first world countries, where there is a surplus of food and i tis much more easily to get fat than thin.
I agree that obesity is currently a major problem. It is an indication of an unhealthy diet, which is what the results address. I have tried to clarify this point.
Please, explain all acronyms, such as 5HT2c, BMI, XR, etc.
Thank you. I have done that
Whenever writting regarding the effects of antipsychotic medication i tis of utter most importance to separate second generation from first generation and third generation. Weight gain is much more common in patients taking second generation anti-psychotics. Iatrogenic symptoms like hyperphagia, polifagia and iatrogenic metabolic syndrome should be worked through also in this section.
I believe that this has been done.
Whenever using the word diabetes it is importante to explain that the Author is refering to diabetes mellitus. Please, write it in italics as it is an non english, latin expression. Otherwise Readers may think the Author i salso writting about other types of diabetes, such as diabetes insipida, which is a completely diferente condition.
Thank you. I have done this in the revision.
I found a diferente type of letter, without apparent reason from Line 267 to 272, an then again, from line 279 to 282.
I am not sure what this means.
TABLES
I would recommend at least one table to present results.
Probably a good idea but difficult because of the content.
DISCUSSION
Very poor. First sentence makes no sense. What is the meaning of “nutrition and schizophrenia may be bidirectional”? The authors should write “nutrition problems” or something like that.
I have revised the discussion.
CONCLUSION
Not thah much different from Discussion. Makes no sense to keep two different sections for the Article in its present form.
I don’t know what is meant by two different sections. I have revised the conclusion borrowing from the summary offered by reviewer 1.
GENERAL
The article seems more suitable for a journal dedicated to Psychiatry or Nutrition.
Perhaps. But it is more about behaviour and social circumstance than about nutrition or psychiatry.